# Cardiovascular magnetic resonance in emergency patients with multivessel disease or unobstructed coronary arteries: a cost-effectiveness analysis in the UK

Elizabeth A Stokes,[1] Brett Doble,[1] Maria Pufulete,[2,3] Barnaby C Reeves,[2,3] Chiara Bucciarelli-Ducci,[3,4] Stephen Dorman,[4] John P Greenwood,[5] Richard A Anderson,[6] Sarah Wordsworth[1]

For numbered affiliations see end of article.

**Correspondence to**
Dr Elizabeth A Stokes;
elizabeth.stokes@ndph.ox.ac.uk

## ABSTRACT

**Objective** To identify the key drivers of cost-effectiveness for cardiovascular magnetic resonance (CMR) when patients activate the primary percutaneous coronary intervention (PPCI) pathway.

**Design** Economic decision models for two patient subgroups populated from secondary sources, each with a 1 year time horizon from the perspective of the National Health Service (NHS) and personal social services in the UK.

**Setting** Usual care (with or without CMR) in the NHS.

**Participants** Patients who activated the PPCI pathway, and for Model 1: underwent an emergency coronary angiogram and PPCI, and were found to have multivessel coronary artery disease. For Model 2: underwent an emergency coronary angiogram and were found to have unobstructed coronary arteries.

**Interventions** Model 1 (multivessel disease) compared two different ischaemia testing methods, CMR or fractional flow reserve (FFR), versus stress echocardiography. Model 2 (unobstructed arteries) compared CMR with standard echocardiography versus standard echocardiography alone.

**Main outcome measures** Key drivers of cost-effectiveness for CMR, incremental costs and quality-adjusted life years (QALYs) and incremental cost-effectiveness ratios.

**Results** In both models, the incremental costs and QALYs between CMR (or FFR, Model 1) versus no CMR (stress echocardiography, Model 1 and standard echocardiography, Model 2) were small (CMR: −£64 (95% CI −£232 to £187)/FFR: £360 (95% CI −£116 to £844) and CMR/FFR: 0.0012 QALYs (95% CI −0.0076 to 0.0093)) and (£98 (95% CI −£199 to £488) and 0.0005 QALYs (95% CI −0.0050 to 0.0077)), respectively. The diagnostic accuracy of the tests was the key driver of cost-effectiveness for both patient groups.

**Conclusions** If CMR were introduced for all subgroups of patients who activate the PPCI pathway, it is likely that diagnostic accuracy would be a key determinant of its cost-effectiveness. Further research is needed to definitively answer whether revascularisation guided by

## Strengths and limitations of this study

► We present the first cost-effectiveness analysis of cardiovascular magnetic resonance (CMR) and fractional flow reserve (FFR) ischaemia tests compared with stress echocardiography in patients who activate the primary percutaneous coronary intervention (PPCI) pathway and are found to have multivessel coronary disease, and the first cost-effectiveness analysis of introducing CMR in patients who activate the PPCI pathway and are found to have unobstructed coronary arteries.

► This work draws on all available evidence in this field to provide guidance on the key drivers of cost-effectiveness for future research.

► The majority of model parameter estimates were based on single studies with small sample sizes, conducted outside the UK where patient pathways differ.

► CMR and FFR tests were treated as reference standards (assumed 100% sensitivity and specificity) in base-case analyses; sensitivity analyses show that relative cost-effectiveness varies substantially depending on assumptions about these parameters.

CMR or FFR leads to different clinical outcomes in acute coronary syndrome patients with multivessel disease.

## INTRODUCTION

Patients with acute coronary syndrome (ACS) and suspected acute thrombotic coronary occlusion activate the primary percutaneous coronary intervention (PPCI) pathway. In the UK National Health Service (NHS), PPCI is the main clinical approach to restore blood flow in the infarcted artery after ST-elevation myocardial infarction (STEMI).[1] Many patients presenting to hospital with STEMI have multivessel disease (40%–65%),[2–7]

which may require secondary revascularisation of the patients' non-infarcted territories. In addition, 3%–16% of patients who activate the PPCI pathway do not receive PPCI because angiography sometimes shows that the coronary arteries are unobstructed.[8][9] These factors result in considerable uncertainty and sometimes a lack of a definitive diagnosis of myocardial infarction (MI) in these patients, which can lead to inappropriate clinical management,[10][11] which may in turn be associated with poor prognosis or unnecessary resource use.[12]

Cardiovascular magnetic resonance (CMR) is a non-invasive imaging technique that assesses heart structure with high temporal resolution and can help inform management decisions for patients with ACS who activate the PPCI pathway. CMR can facilitate differential diagnosis in the context of a normal coronary angiogram,[13–15] providing a definitive diagnosis in 65%–90% of patients.[13][16][17] The benefits of CMR in other subgroups of PPCI patients, such as those found to have multivessel disease, are less clear. Long-term outcome data and studies reporting the cost-effectiveness of CMR are absent, further complicating resource allocation of CMR services within the NHS. Despite a lack of evidence, CMR has been increasingly used in patients with ACS in the UK[18] and is included in the 2017 European Society of Cardiology guidelines on the management of patients presenting with STEMI.[19]

There has also been a rapid increase in the use of fractional flow reserve (FFR) testing: a guide wire-based procedure which is undertaken through a standard guide catheter during invasive angiography to assess the degree of ischaemia in stenosed coronary arteries. Stress CMR is considered to have excellent diagnostic accuracy compared with FFR for detecting ischaemia.[20][21] The main benefit of ischaemia testing using FFR is that lesions can be assessed and revascularised in the same session; however, if revascularisation is not required, then the benefit of FFR over CMR is less clear. In addition to CMR and FFR, stress echocardiography (ECHO) and myocardial perfusion scintigraphy using single photon emission CT are often used to evaluate patients with multivessel disease for residual ischaemia of bystander disease after PPCI for STEMI; stress ECHO is one of the most commonly used imaging modalities in the UK.[22]

In this paper, we present a health economic analysis based on a study that was designed to establish the feasibility of setting up a UK multicentre registry to document CMR use in patients who activate the PPCI pathway and determine its prognostic value and impact on patient management.[23] Here, we present the results of two economic decision models which we designed to compare the cost-effectiveness of introducing CMR in two patient subgroups: patients with multivessel disease and those with unobstructed coronary arteries. The aim was to identify how the results of the analyses change in response to sensitivity analyses to determine key drivers of cost-effectiveness that could be considered for detailed measurement in any future research.

## METHODS

### Overview and treatment strategies

We built two separate cost-effectiveness models in consultation with clinical experts to estimate the healthcare costs and quality-adjusted life years (QALYs) of introducing CMR after PPCI pathway activation in two patient subgroups over a 1 year time horizon:

1. Model 1 compares two types of ischaemia testing: CMR or FFR to stress ECHO in patients with multivessel disease (commonly defined as stenosis >50%) after index angiogram and PPCI.
2. Model 2 compares CMR in addition to standard ECHO to standard ECHO alone in patients with unobstructed coronary arteries after index angiogram.

Given the focus on the immediate consequences on clinical management of introducing additional diagnostic testing, and because this is a feasibility study assessing key drivers of cost-effectiveness rather than cost-effectiveness per se, we chose a 1 year time horizon. Survival to 12 months under each testing strategy was assessed to determine whether there were differences, which would identify a need to consider a longer time horizon. From these two models, it may be possible to make inferences about the likely drivers of cost-effectiveness of introducing CMR for all subgroups of patients who activate the PPCI pathway. The analysis was conducted from an NHS and personal social services perspective in the UK.[24]

### Model structure

#### Model 1: multivessel disease

Model 1 uses a decision tree structure (figure 1) to compare the cost-effectiveness of two types of ischaemia testing (CMR and FFR) to stress ECHO. All three tests are used to guide decisions regarding secondary revascularisation in patients with multivessel disease. Based on the findings of their emergency index angiogram at which the culprit lesion is treated, patients can enter one of three test/treatment pathways to determine whether further treatment is required for non-culprit lesions at a later date: no ischaemia testing and secondary revascularisation; ischaemia testing to guide decisions for secondary revascularisation or no ischaemia testing and no secondary revascularisation. Patients are then divided into whether they 'truly' have ischaemia or not. For patients receiving ischaemia testing, the test results can be either positive or negative (ischaemia present or not), and the model reflects potential misclassification of patients (ie, ischaemia testing has a sensitivity and specificity). Patients entering any of the three test/treatment pathways are then at risk of experiencing a major adverse cardiovascular event (MACE) or no MACE over the 1 year time horizon. MACE is a composite endpoint of all-cause mortality, MI, stroke or (repeat) revascularisation.

The model structure is the same for the three ischaemia testing options, but there are differences in the timing of testing and secondary revascularisation between these arms (note for all arms this testing and secondary revascularisation, if required, is at a later date than the index

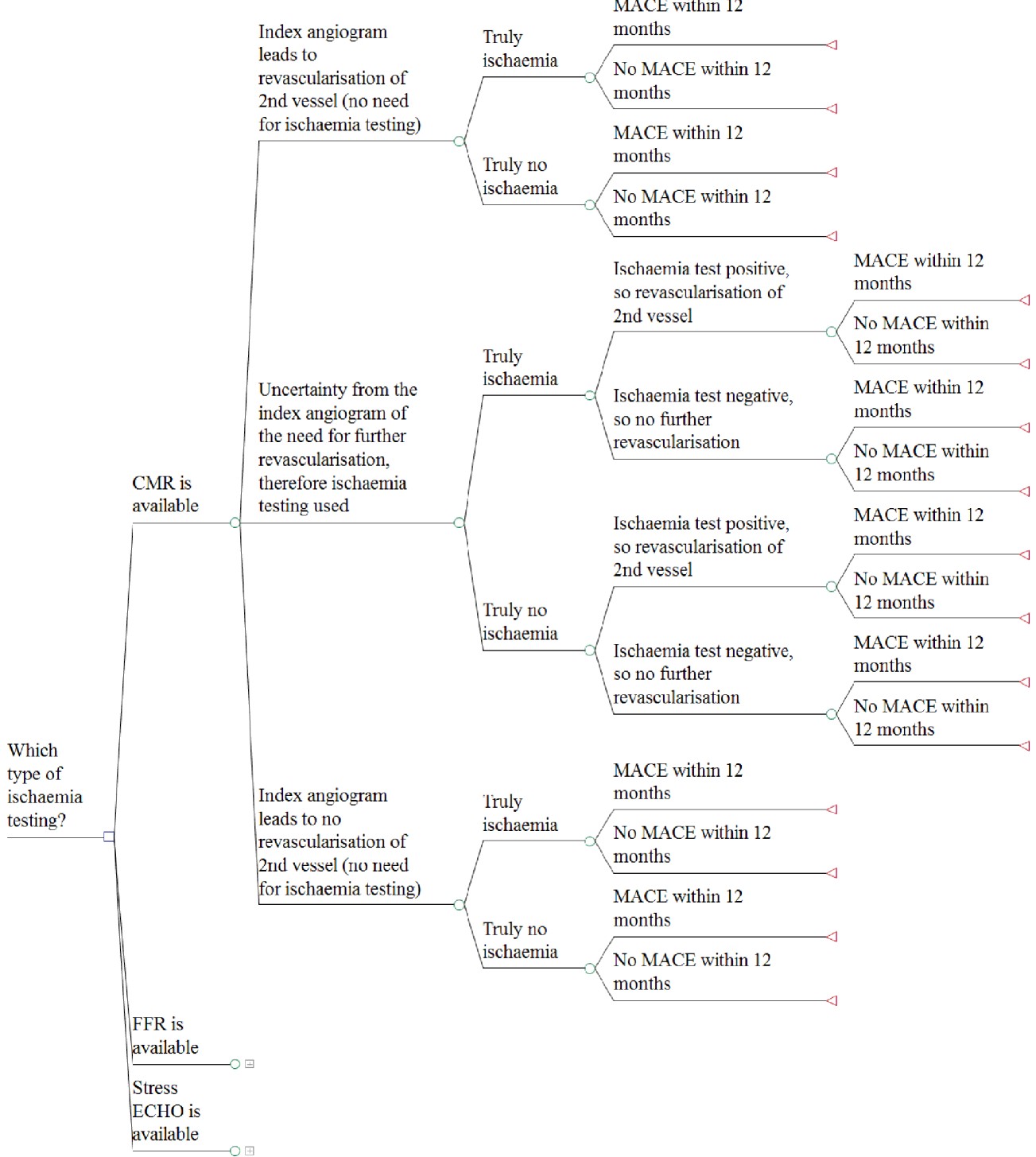

**Figure 1** Model 1 structure—patients with multivessel disease. Patient pathways for FFR and stress ECHO are identical to those for CMR. CMR, cardiovascular magnetic resonance; ECHO, echocardiography; FFR; fractional flow reserve; MACE, major adverse cardiovascular events.

angiogram at PPCI and treatment of the culprit lesion). For CMR and stress ECHO, this is a two-step process (ie, a patient has the ischaemia test and then separately has secondary revascularisation). For FFR, this happens at the same time (ie, a patient has their diagnostic angiogram and FFR, and if required, secondary revascularisation). We have assumed that this timing difference only affects costs, but not the model structure.

### Model 2: unobstructed coronary arteries

Model 2 also uses a decision tree structure (figure 2) to compare the cost-effectiveness of CMR with standard ECHO to 'current practice' of standard ECHO only. These tests are used to guide decisions in offering treatment for MI in patients who activate the PPCI pathway but are found to have unobstructed coronary arteries. In each arm, patients are divided according to whether they truly

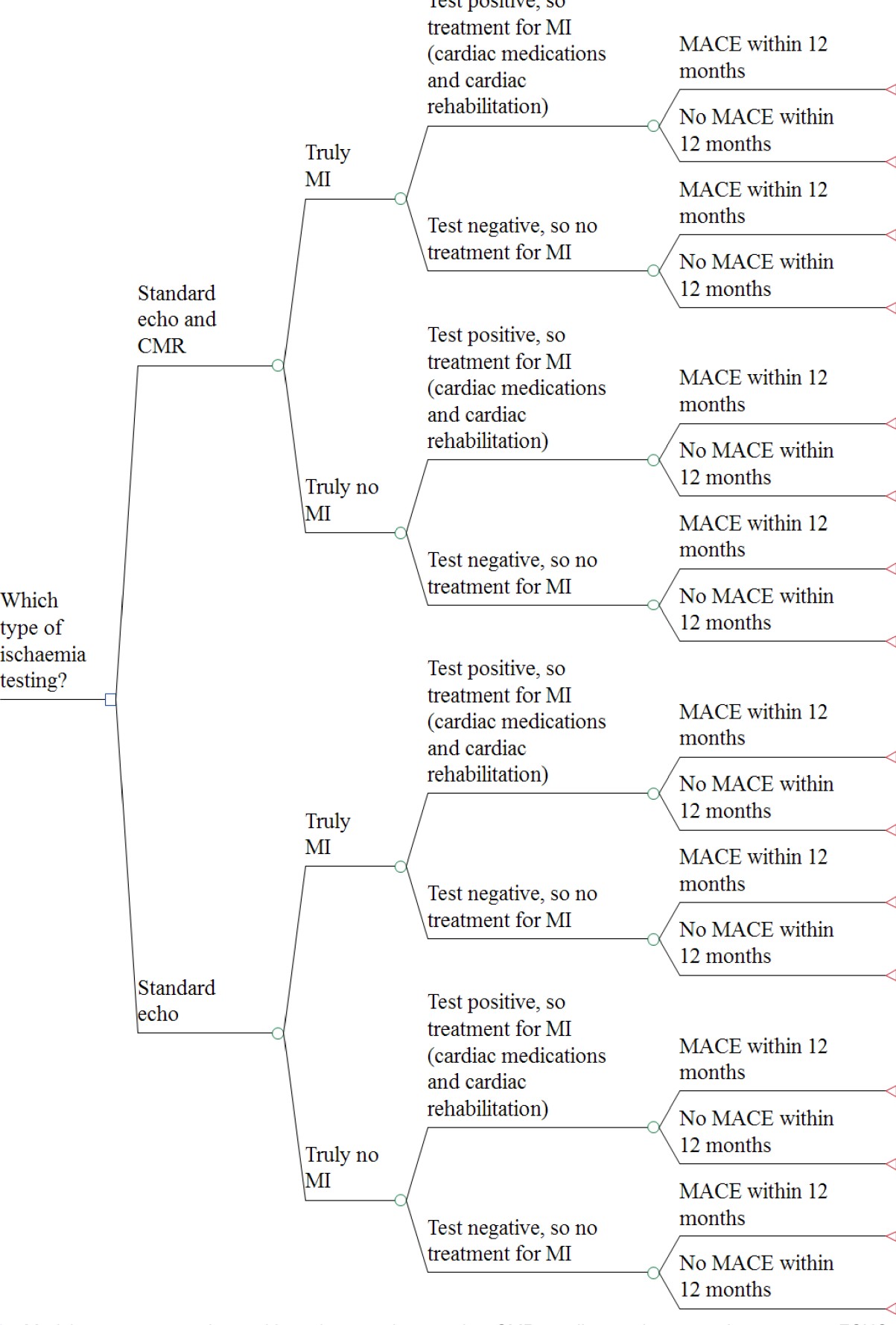

**Figure 2** Model 2 structure—patients with unobstructed coronaries. CMR, cardiovascular magnetic resonance; ECHO, echocardiography; MACE, major adverse cardiovascular events; MI, myocardial infarction.

had an MI or not. Results for both testing approaches can be either positive or negative (MI occurred or not), with the model again reflecting the potential for misclassification. Patients receiving a positive test result are assumed to have had an MI and receive treatment (cardiac rehabilitation and medications). A negative test results in treatment for other cardiac causes of chest pain (fewer cardiac medications or none at all). Thereafter, patients are divided into whether they had MACE or not over the 1 year time horizon.

## Economic analyses

TreeAge Pro 2013 (TreeAge Software, Williamstown, Massachusetts, USA) was used to develop and analyse the models. Resources were valued in 2015/2016 pounds sterling. The expected costs and QALYs associated with each testing strategy to 1 year were calculated. Costs and QALYs were not discounted, as the time horizon was only 1 year. Incremental cost-effectiveness ratios (ICERs) were calculated as the ratio of the difference in cost to the difference in QALYs between each testing option and its comparator.

## Identification of model parameters

Testing and clinical parameters used in both models were derived from reviews of the literature. Unit costs were obtained from standard UK sources, notably the National Schedule of Reference Costs and the British National Formulary.[25 26] Health state utility values were identified by searching the Tufts Cost-effectiveness Analysis (CEA) Registry.[27]

For model 1, we searched Medline using Medical Subject Headings and keywords for each ischaemia test (eg, exp/pressure-wire or pressure-wire.mp or fractional flow reserve.mp and exp/myocardial infarction). Review articles, reference lists and citations of relevant studies were also evaluated. For model 2, initial Medline searching identified a 2016 review paper,[28] which referenced the inaugural study of CMR in patients with unobstructed coronary arteries.[13] All citations of this primary paper were reviewed.[13] Two meta-analyses were identified and their citations were reviewed.[29 30] Medline was also searched using keywords such as 'myocardial infarction with non-obstructive coronary arteries' and synonyms.

Complete lists of model parameters, their respective sources and assumptions made in their estimation are provided in online supplementary appendices A and B and briefly summarised below.

## Testing parameters
### Model 1: multivessel disease

CMR and FFR were treated as reference standards with assumed 100% sensitivity and specificity. While FFR is the reference standard for detecting ischaemia in patients with stable coronary artery disease, there is uncertainty in how FFR-guided revascularisation influences outcome in this high-risk group of STEMI patients with multivessel disease. It was therefore considered a reasonable

base-case assumption that the sensitivities and specificities of CMR and FFR were the same in this population. While in practice their sensitivity and specificity are likely to be a little less than 100%, there is no gold standard test for ischaemia in STEMI patients with multivessel disease with which to compare; lower sensitivities and specificities were explored in sensitivity analyses. The assumption of 100% sensitivity and specificity for CMR and FFR results in identical probabilities and outcomes, but differences in costs for the two arms. Sensitivity and specificity values for stress ECHO were obtained from Gurunathan et al.[31]

### Model 2: unobstructed coronary arteries

CMR with standard ECHO was treated as a reference standard, whereas sensitivity and specificity values for standard ECHO were obtained from Dastidar et al.[32]

## Clinical parameters
### Model 1: multivessel disease

The probabilities of patients entering one of the three test/treatment pathways were based on expert clinical opinion. Probabilities of true ischaemia for each test/treatment pathway were derived from the FAME study.[33] Subgroup outcomes from Smits et al were used to estimate probabilities of MACE for three patient groups: truly ischaemic patients who had secondary revascularisation, truly ischaemic patients who did not have secondary revascularisation and truly non-ischaemic patients, regardless of secondary revascularisation.[34]

### Model 2: unobstructed coronary arteries

Studies reporting the probability of MACE in patients with unobstructed coronary arteries have used varying definitions of MACE and follow-up periods (12–34 months).[35–38] In patients with <50% coronary artery stenosis, Kang et al reported a 7.8% risk of MACE at 1 year,[37] but their MACE definition did not include stroke. Three studies reported MACE, broken down by CMR diagnosis.[35 36 38] Pathik et al reported MACE for 27% of patients diagnosed with MI, and for 5% of patients without MI (diagnosed with myocarditis, cardiomyopathy or normal CMR) over a median of 24 months.[38] In the other two studies, no MACE outcomes were observed in the patients without MI. The MACE estimates reported in Pathik et al were converted to continuous rates (assuming events occur evenly over 24 months of follow-up). The ratio of events in the patients with MI compared with the patients without MI was 6 and was used with the overall estimate of MACE at 1 year from Kang et al (0.078),[37] and the probability of patients truly having an MI from Pasupathy et al (0.24),[29] to calculate the probability of MACE in patients who truly had an MI (0.21) and in those who did not have an MI (0.03).

The hazard ratio for MACE in patients who were taking statins compared with those who were not (0.77) reported by Lindahl et al was used to adjust the probability of MACE for patients who had an MI, giving a probability of MACE for those with MI but without treatment of 0.26.[39]

For patients without MI, the probability of MACE of 0.03 was assumed to apply regardless of treatment.

## Cost parameters

### Model 1: multivessel disease

The costs considered for all three ischaemia testing strategies included: the cost of ischaemia testing (CMR, FFR or stress ECHO) if required, secondary revascularisation, adverse events included in MACE (initial inpatient and postdischarge costs), medications and cardiac rehabilitation offered to all patients and additional healthcare costs beyond hospital discharge to 1 year, including follow-up outpatient appointments.

### Model 2: unobstructed coronary arteries

The costs considered for the two testing strategies included: the cost of testing (CMR and/or standard ECHO), cardiac rehabilitation and medications for patients classified as either having had an MI or a non-MI diagnosis (details in online supplementary appendix B), adverse events included in MACE (initial inpatient and postdischarge costs) and additional healthcare costs beyond hospital discharge to 1 year, including a follow-up appointment.

## Quality-adjusted life years

### Model 1: multivessel disease

QALYs were estimated over 1 year as a function of survival and health state utility values. Since death is one of the components of MACE, we assumed that patients without MACE are all alive at 1 year and that the probability of dying for MACE patients was 0.08 as estimated by Smits *et al.*[34] We identified mean utility values of 0.72, 0.77 and 0.77 at baseline (during index hospitalisation), 1 and 12 months, respectively, estimated using the UK EQ-5D-3L tariff in patients with STEMI having PPCI.[27 40]

QALYs were calculated assuming a patient's utility changed linearly between each time point or death. Patients who died were assumed to die midway through the time-period (6 months) and have a utility of zero after death. QALYs for patients without MACE who were all alive at 1 year were calculated based on the utility values for the three time-points above.[40] QALYs for patients with MACE were also calculated using these three utility values, but assuming 8% of patients died to produce a weighted average for patients alive and dead at 1 year.[34] A utility decrement of 0.05 was applied to this weighted average to reflect the reduced quality-of-life of patients with MACE compared with those without.[41] QALYs for patients who had secondary revascularisation were modified to assume their utility at the 1 month time-point was repeated for a subsequent month.

### Model 2: unobstructed coronary arteries

Searches of the CEA Registry did not yield any utility values for patients with unobstructed coronary arteries, and therefore QALYs for patients with and without MACE in model 2 were calculated using the same utility values and methods as for model 1.[40]

## Base-case and sensitivity analyses

The best available point estimates were used in the base-case analyses in which ICERs were the summary measures. Since this was a feasibility study, the actual ICERs are not the main interest, but rather how they change in response to sensitivity analyses.

One-way deterministic sensitivity analyses (DSAs) were conducted to test the impact of varying the base-case values for several model parameters. Alternative values for DSAs (online supplementary appendices C and D) were obtained from substitute evidence sources, clinical judgement (eg, 8% probability of death for patients with MACE was thought to be low, and a relative risk of 6 to be high for events in patients with MI compared with patients without MI, so higher and lower values were explored) or altering mean values for unit costs by ±20%. Threshold analyses were also conducted to establish the cost of each test at which the expected costs associated with an alternative strategy would be identical.

Probabilistic sensitivity analyses (PSAs) were used to account for the impact on results of all uncertain parameters simultaneously. Parameters were assigned distributions based on the precision of estimates available to describe a range of plausible values, and 1000 randomly selected values from each distribution were generated and results calculated for each run of the model. 95% CI and scatter plots of the 1000 incremental costs and QALYs for each comparison were then assessed.

## Patient and public involvement

Patients and the public were not involved in the design of this study.

## RESULTS

### Base-case results

#### Model 1: multivessel disease

Over 1 year, the expected total costs of stress ECHO, CMR and FFR strategies per patient were £5495, £5431 and £5855, respectively (table 1). Use of stress ECHO produced 0.7564 QALYs, whereas CMR and FFR both resulted in an additional 0.0012 QALYs gained. CMR dominated stress ECHO as a CMR strategy was less costly (–£64) and more effective (+0.0012 QALYs). FFR was more costly (£360) and more effective (+0.0012 QALYs) than stress ECHO, resulting in an ICER above the accepted cost-effectiveness threshold of £20 000–£30 000 per QALY.

#### Model 2: unobstructed coronary arteries

Over 1 year, the expected total costs of CMR with standard ECHO and standard ECHO alone strategies per patient were £3130 and £3032, respectively (table 2). Use of CMR with standard ECHO produced 0.0005 more QALYs than standard ECHO alone, at an increase in costs of £98, resulting in an ICER above the accepted cost-effectiveness threshold.

**Table 1** Base-case and deterministic one-way sensitivity analysis results for model 1 (multivessel disease)

| DSA | Ischaemia testing option | Costs (£) Mean (SE) | QALYs Mean (SE) | Difference in costs to Stress ECHO Mean (95% CI) | Difference in QALYs to Stress ECHO Mean (95% CI) | ICER (£)—compared with stress ECHO |
|---|---|---|---|---|---|---|
| Base case | Stress ECHO | 5495 (556) | 0.7564 (0.0545) | | | |
| | CMR | 5431 (560) | 0.7576 (0.0551) | −64 (−232 to 187) | 0.0012 (−0.0076 to 0.0093) | CMR dominant (−53 563) |
| | FFR | 5855 (539) | 0.7576 (0.0551) | 360 (−116 to 844) | 0.0012 (−0.0076 to 0.0093) | 300216 |
| 1 Decision from angiogram | Stress ECHO | 4575 (449) | 0.7537 (0.0654) | | | |
| | CMR | 4558 (500) | 0.7540 (0.0661) | −17 (−62 to 45) | 0.0003 (−0.0022 to 0.0026) | CMR dominant (−53 563) |
| | FFR | 4667 (500) | 0.7540 (0.0661) | 93 (−21 to 214) | 0.0003 (−0.0022 to 0.0026) | 300216 |
| 2a CMR test sensitivity | Stress ECHO | 5495 (556) | 0.7564 (0.0545) | | | |
| | CMR | 5349 (549) | 0.7569 (0.0553) | −146 (−362 to 105) | 0.0005 (−0.0162 to 0.0181) | CMR dominant (−276 922) |
| 2b CMR test specificity | Stress ECHO | 5495 (556) | 0.7564 (0.0545) | | | |
| | CMR | 5639 (593) | 0.7574 (0.0554) | 145 (−103 to 524) | 0.0010 (−0.0155 to 0.0192) | 138608 |
| 2c CMR test accuracy | Stress ECHO | 5495 (556) | 0.7564 (0.0545) | | | |
| | CMR | 5558 (600) | 0.7567 (0.0572) | 63 (−202 to 390) | 0.0004 (−0.0119 to 0.0108) | 168453 |
| 3a FFR test sensitivity | Stress ECHO | 5495 (556) | 0.7564 (0.0545) | | | |
| | FFR | 5817 (555) | 0.7569 (0.0564) | 322 (−138 to 782) | 0.0005 (−0.0162 to 0.0181) | 610864 |
| 3b FFR test specificity | Stress ECHO | 5495 (556) | 0.7564 (0.0545) | | | |
| | FFR | 5982 (567) | 0.7574 (0.0572) | 487 (13 to 1024) | 0.0010 (−0.0155 to 0.0192) | 466754 |
| 3c FFR test accuracy | Stress ECHO | 5495 (556) | 0.7564 (0.0545) | | | |
| | FFR | 5944 (568) | 0.7567 (0.0572) | 449 (−18 to 947) | 0.0004 (−0.0119 to 0.0108) | 1207875 |
| 4a ECHO test sensitivity | Stress ECHO | 5534 (552) | 0.7567 (0.0562) | | | |
| | CMR | 5431 (560) | 0.7576 (0.0551) | −103 (−274 to 124) | 0.0009 (−0.0122 to 0.0114) | CMR dominant (−117 499) |
| | FFR | 5855 (539) | 0.7576 (0.0551) | 321 (−157 to 761) | 0.0009 (−0.0122 to 0.0114) | 365070 |
| 4b ECHO test specificity | Stress ECHO | 5392 (565) | 0.7564 (0.0562) | | | |
| | CMR | 5431 (560) | 0.7576 (0.0551) | 39 (−105 to 294) | 0.0011 (−0.0035 to 0.0067) | 34 961 |
| | FFR | 5855 (539) | 0.7576 (0.0551) | 463 (54 to 935) | 0.0011 (−0.0035 to 0.0067) | 413065 |
| 4c ECHO test accuracy | Stress ECHO | 5431 (553) | 0.7568 (0.0555) | | | |
| | CMR | 5431 (560) | 0.7576 (0.0551) | +0 (−128 to 253) | 0.0008 (−0.0022 to 0.0036) | 201 |
| | FFR | 5855 (539) | 0.7576 (0.0551) | 424 (18 to 888) | 0.0008 (−0.0022 to 0.0036) | 529193 |
| 5 MACE +0.05 | Stress ECHO | 5674 (586) | 0.7520 (0.0566) | | | |
| | CMR | 5610 (580) | 0.7533 (0.0574) | −65 (−233 to 164) | 0.0012 (−0.0078 to 0.0087) | CMR dominant (−53 434) |
| | FFR | 6033 (575) | 0.7533 (0.0574) | 359 (−85 to 831) | 0.0012 (−0.0078 to 0.0087) | 296587 |
| 5 MACE −0.05 | Stress ECHO | 5317 (564) | 0.7607 (0.0589) | | | |
| | CMR | 5252 (565) | 0.7619 (0.0595) | −65 (−244 to 192) | 0.0012 (−0.0081 to 0.0097) | CMR dominant (−53 463) |
| | FFR | 5676 (533) | 0.7619 (0.0595) | 359 (−112 to 830) | 0.0012 (−0.0081 to 0.0097) | 297408 |

Continued

**Table 1** Continued

| DSA | Ischaemia testing option | Costs (£) Mean (SE) | QALYs Mean (SE) | Difference in costs to Stress ECHO Mean (95% CI) | Difference in QALYs to Stress ECHO Mean (95% CI) | ICER (£) – compared with stress ECHO |
|---|---|---|---|---|---|---|
| 6 CMR cost +20% | Stress ECHO | 5495 (556) | 0.7564 (0.0545) | | | |
| | CMR | 5463 (556) | 0.7576 (0.0551) | −32 (−218 to 187) | 0.0012 (−0.0076 to 0.0093) | CMR dominant (−27 021) |
| 6 CMR cost −20% | Stress ECHO | 5495 (556) | 0.7564 (0.0545) | | | |
| | CMR | 5399 (551) | 0.7576 (0.0551) | −96 (−268 to 159) | 0.0012 (−0.0076 to 0.0093) | CMR dominant (−80 104) |
| 7 FFR cost +20% | Stress ECHO | 5495 (556) | 0.7564 (0.0545) | | | |
| | FFR | 6015 (560) | 0.7576 (0.0551) | 521 (78 to 992) | 0.0012 (−0.0076 to 0.0093) | 434 425 |
| 7 FFR cost −20% | Stress ECHO | 5495 (556) | 0.7564 (0.0545) | | | |
| | FFR | 5694 (537) | 0.7576 (0.0551) | 199 (−240 to 622) | 0.0012 (−0.0076 to 0.0093) | 166 007 |
| 8 QALYs for MACE −0.2 | Stress ECHO | 5495 (556) | 0.7306 (0.0559) | | | |
| | CMR | 5431 (560) | 0.7346 (0.0565) | −64 (−232 to 187) | 0.0040 (−0.0057 to 0.0117) | CMR dominant (−16 041) |
| | FFR | 5855 (539) | 0.7346 (0.0565) | 360 (−116 to 844) | 0.0040 (−0.0057 to 0.0117) | 89 907 |
| 9 QALYs for no MACE +0.2 | Stress ECHO | 5495 (556) | 0.9306 (0.0576) | | | |
| | CMR | 5431 (560) | 0.9346 (0.0587) | −64 (−232 to 187) | 0.0040 (−0.0071 to 0.0104) | CMR dominant (−16 041) |
| | FFR | 5855 (539) | 0.9346 (0.0587) | 360 (−116 to 844) | 0.0040 (−0.0071 to 0.0104) | 89 907 |

Note: all costs and ICERs are rounded to nearest pound.
CMR, cardiovascular magnetic resonance; DSAs, deterministic sensitivity analyses; ECHO, echocardiography; FFR, fractional flow reserve; ICER, incremental cost-effectiveness ratio; MACE, major adverse cardiovascular events; QALY, quality-adjusted life year.

**Table 2** Base-case and deterministic one-way sensitivity analysis results for model 2 (unobstructed coronary arteries)

| DSA | Testing option | Costs (£) Mean (SE) | QALYs Mean (SE) | Difference in costs Mean (95% CI) | Difference in QALYs Mean (95% CI) | ICER (£) |
|---|---|---|---|---|---|---|
| Base case | Standard ECHO | 3032 (564) | 0.7615 (0.0837) | | | |
| | Standard ECHO and CMR | 3130 (589) | 0.7620 (0.0844) | 98 (−199 to 488) | 0.0005 (−0.0050 to 0.0077) | 190114 |
| 1a CMR test sensitivity | Standard ECHO | 3032 (564) | 0.7615 (0.0837) | | | |
| | Standard ECHO and CMR | 3121 (596) | 0.7618 (0.0838) | 89 (−163 to 460) | 0.0003 (−0.0030 to 0.0048) | 278043 |
| 1b CMR test specificity | Standard ECHO | 3032 (564) | 0.7615 (0.0837) | | | |
| | Standard ECHO and CMR | 3191 (612) | 0.7620 (0.0840) | 160 (−132 to 567) | 0.0005 (−0.0047 to 0.0076) | 309400 |
| 1c CMR test accuracy | Standard ECHO | 3032 (564) | 0.7615 (0.0837) | | | |
| | Standard ECHO and CMR | 3183 (591) | 0.7618 (0.0850) | 151 (−105 to 523) | 0.0003 (−0.0026 to 0.0053) | 469998 |
| 2a ECHO test sensitivity | Standard ECHO | 3042 (569) | 0.7617 (0.0835) | | | |
| | Standard ECHO and CMR | 3130 (589) | 0.7620 (0.0844) | 88 (−163 to 479) | 0.0003 (−0.0029 to 0.0042) | 299955 |
| 2b ECHO test specificity | Standard ECHO | 2935 (539) | 0.7615 (0.0824) | | | |
| | Standard ECHO and CMR | 3130 (589) | 0.7620 (0.0844) | 195 (−95 to 600) | 0.0005 (−0.0047 to 0.0070) | 378523 |
| 2c ECHO test accuracy | Standard ECHO | 2945 (561) | 0.7617 (0.0863) | | | |
| | Standard ECHO and CMR | 3130 (589) | 0.7620 (0.0844) | 185 (−59 to 581) | 0.0003 (−0.0027 to 0.0042) | 631744 |
| 3 MACE (ratio=4) | Standard ECHO | 3067 (541) | 0.7608 (0.0826) | | | |
| | Standard ECHO and CMR | 3165 (559) | 0.7614 (0.0830) | 98 (−192 to 465) | 0.0005 (−0.0043 to 0.0069) | 190154 |
| 4 MACE (ratio=2) | Standard ECHO | 3043 (573) | 0.7613 (0.0829) | | | |
| | Standard ECHO and CMR | 3146 (606) | 0.7617 (0.0834) | 104 (−162 to 498) | 0.0004 (−0.0051 to 0.0063) | 250905 |
| 5 MACE (ratio=1) | Standard ECHO | 3046 (579) | 0.7612 (0.0847) | | | |
| | Standard ECHO and CMR | 3161 (596) | 0.7614 (0.0850) | 115 (−168 to 492) | 0.0002 (−0.0051 to 0.0069) | 554609 |
| 6 CMR cost +20% | Standard ECHO | 3032 (564) | 0.7615 (0.0837) | | | |
| | Standard ECHO and CMR | 3183 (579) | 0.7620 (0.0844) | 151 (−133 to 532) | 0.0005 (−0.0050 to 0.0077) | 292836 |
| 6 CMR cost −20% | Standard ECHO | 3032 (564) | 0.7615 (0.0837) | | | |
| | Standard ECHO and CMR | 3077 (572) | 0.7620 (0.0844) | 45 (−229 to 476) | 0.0005 (−0.0050 to 0.0077) | 87391 |
| 7 Reduce MACE costs | Standard ECHO | 2974 (548) | 0.7615 (0.0837) | | | |
| | Standard ECHO and CMR | 3077 (574) | 0.7620 (0.0844) | 103 (−159 to 502) | 0.0005 (−0.0050 to 0.0077) | 198992 |
| 8 QALYs for MACE −0.2 | Standard ECHO | 3032 (564) | 0.7457 (0.0879) | | | |
| | Standard ECHO and CMR | 3130 (589) | 0.7474 (0.0886) | 98 (−199 to 488) | 0.0018 (−0.0091 to 0.0139) | 55281 |
| 9 QALYs for no MACE +0.2 | Standard ECHO | 3032 (564) | 0.9457 (0.0905) | | | |
| | Standard ECHO and CMR | 3130 (589) | 0.9474 (0.0911) | 98 (−199 to 488) | 0.0018 (−0.0094 to 0.0134) | 55281 |
| 10 QALYs for MACE (50% die) | Standard ECHO | 2994 (516) | 0.7488 (0.0859) | | | |
| | Standard ECHO and CMR | 3095 (542) | 0.7503 (0.0864) | 101 (−180 to 464) | 0.0015 (−0.0090 to 0.0133) | 66040 |

Note: all costs and ICERs are rounded to nearest pound.
CMR, cardiovascular magnetic resonance; DSAs, deterministic sensitivity analyses; ECHO, echocardiography; ICER, incremental cost-effectiveness ratio; MACE, major adverse cardiovascular events; QALY, quality-adjusted life year.

## Sensitivity analyses

### Model 1: multivessel disease

Table 1 reports the results of the DSAs for model 1. DSA 1, varying the probabilities of a clinician ordering an ischaemia test and of a decision to perform a secondary revascularisation (or not), affects the mean costs and QALYs under each strategy but does not alter the cost-effectiveness results. When the specificity, or sensitivity and specificity of CMR were reduced to 80% (DSA 2), CMR became more costly than stress ECHO, rather than dominant, and at accepted cost-effectiveness thresholds, would not be considered cost-effective relative to stress ECHO. When the sensitivity and/or specificity of FFR were reduced to 80% (DSA 3), conclusions did not alter, but the ICER increased. In DSA 4, using higher sensitivity, or sensitivity and specificity values for stress ECHO, the ICER for CMR would still be considered cost-effective but when using a higher specificity only for stress ECHO, CMR would no longer be considered cost-effective. Improved sensitivity and/or specificity for stress ECHO resulted in an even more unfavourable ICER for FFR. Altering the probability of MACE (DSA 5), the costs of the ischaemia tests (DSAs 6 and 7), and QALYs for patients with and without MACE (DSAs 8 and 9), did not alter cost-effectiveness conclusions.

Threshold analyses identified that if the cost of CMR increased to £371 (from a base-case value of £264), there would be no difference in the expected costs of the CMR and stress ECHO strategies. Similarly, the cost of angiogram and FFR would need to be reduced to £250 and £415 (from a base-case value of £1340) for there to be no difference in the expected costs of CMR and stress ECHO strategies, respectively compared with FFR.

The 1000 simulated incremental costs and QALYs from the PSA (figure 3A and B) show considerable uncertainty in the base-case results. This uncertainty is apparent in both figures as the spread of estimates crosses two or three quadrants of the cost-effectiveness plane.

Under all testing strategies, the probability of survival to 12 months was 0.99. If a cohort of 1000 patients entered each strategy, on average there would be one more death by 12 months under a stress ECHO strategy compared with CMR/FFR (11 vs 10 deaths). Given that there is no difference in survival, we did not extrapolate findings to a longer time horizon.

### Model 2: unobstructed coronary arteries

Table 2 reports the results of the DSAs for model 2. Reducing the diagnostic accuracy of CMR (DSA 1), increasing the diagnostic accuracy of standard ECHO (DSA 2) and increasing the cost of CMR (DSA 6) had the greatest impact on the mean incremental cost between CMR with standard ECHO and standard ECHO alone. Altering QALYs (DSAs 8, 9 and 10), had the greatest impact on the mean incremental QALYs between the arms. However, in all DSAs conducted, CMR with standard ECHO remained both more costly and more effective than standard ECHO alone, resulting in ICERs that were consistently above the accepted cost-effectiveness threshold.

Threshold analyses show that if the cost of CMR reduced to £166 (from a base-case value of £264), there would be no difference in the expected costs between strategies.

Similar to model 1, the 1000 simulated incremental costs and QALYs (figure 3C) indicate considerable uncertainty in the base-case results; again, the estimates are spread across three quadrants of the cost-effectiveness plane.

Under both strategies, the probability of survival to 12 months was 0.99. If a cohort of 1000 patients entered each strategy, on average there would be one more death by 12 months under a standard ECHO strategy compared with standard ECHO and CMR (seven vs six deaths). As for model 1, given that there is no difference in survival, we did not extrapolate findings to a longer time horizon.

## DISCUSSION

We developed economic decision models to identify key drivers of cost-effectiveness of CMR compared with 'current practice' in two subgroups of patients who activate the PPCI pathway: multivessel disease (model 1) and unobstructed coronary arteries (model 2). 'Current practice' for ischaemia testing in multivessel disease varies widely. The results of both models suggest that differences in QALYs between strategies are small and, therefore, the results are largely driven by modest differences in costs.

In model 1, only 35% of patients receiving ischaemia testing truly had ischaemia, therefore the majority of patients received an expensive test without needing further treatment. If more patients needed revascularisation, the expected cost of the FFR strategy would reduce as potential economies of scale would result from performing the ischaemia test and secondary revascularisation concurrently. In model 2, the reduction in costs associated with treating fewer patients for MI if CMR were to be introduced only partially compensated for the additional cost of CMR. DSAs for both models identified the diagnostic accuracy of the tests as the key driver of cost-effectiveness. In model 1, the costs of ischaemia testing and QALYs associated with MACE/no MACE also influenced cost-effectiveness, but the latter factors had minimal impact compared with altering the diagnostic accuracy of CMR and FFR. For model 2, the cost of CMR, the QALYs associated with MACE/no MACE, and the proportion of patients with MACE who die, influenced cost-effectiveness results.

Overall, we have identified the sensitivity and specificity of the tests as the key drivers of cost-effectiveness in both models. It is likely that any business case to introduce CMR for all patients who activate the PPCI pathway would require accurate estimates of these parameters to be collected in future studies.

In terms of limitations of our study, first, many parameter estimates were obtained from a single study conducted outside the UK, where practices may differ. Estimates

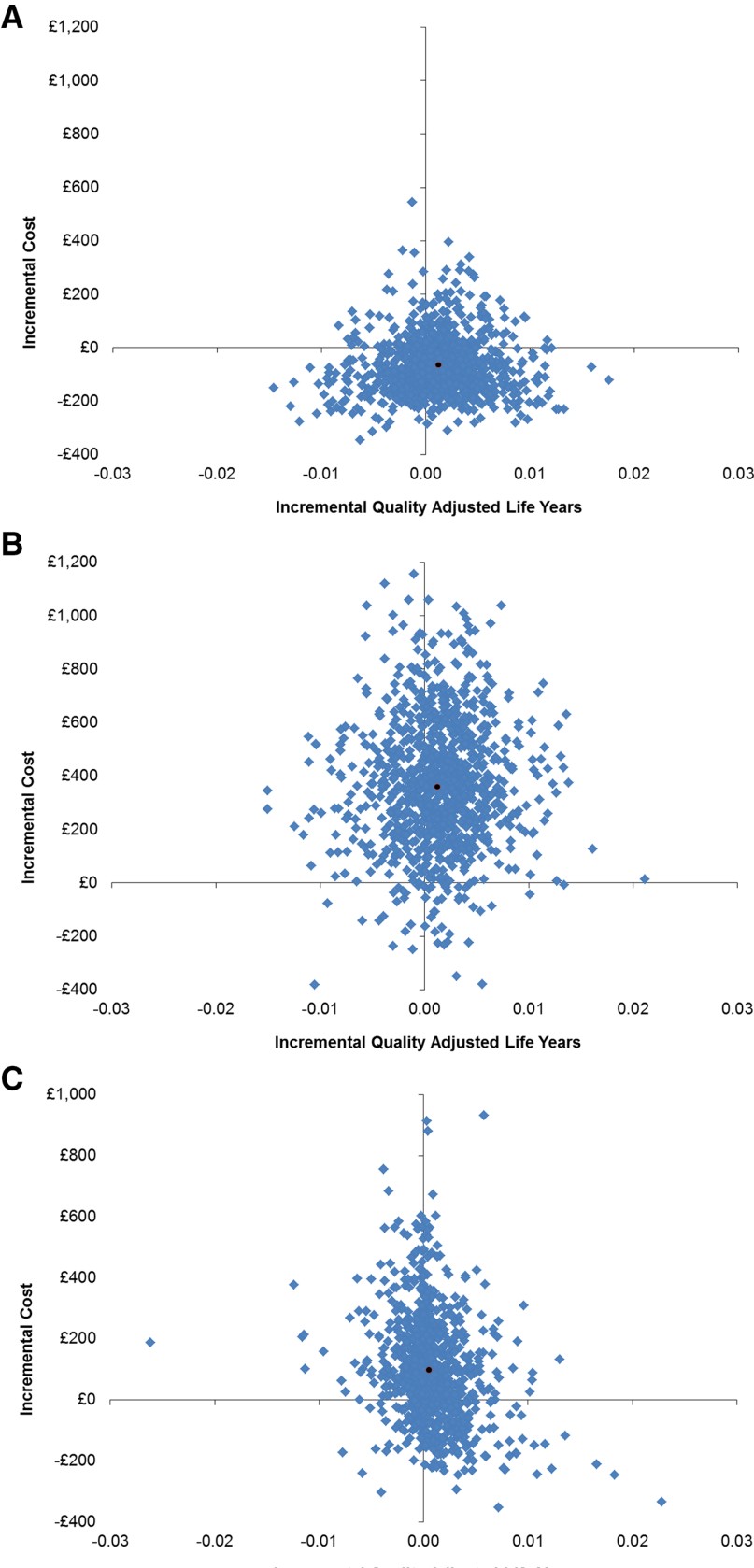

**Figure 3** (A) Model 1: plot of the 1000 simulated cost and QALY differences from the PSA for CMR versus stress ECHO. (B) Model 1: plot of the 1000 simulated cost and QALY differences from the PSA for FFR versus stress ECHO. (C) Model 2: plot of the 1000 simulated cost and QALY differences from the PSA for standard ECHO and CMR versus standard ECHO. CMR, cardiovascular magnetic resonance; ECHO, echocardiography; FFR; fractional flow reserve; PSA, probabilistic sensitivity analyses; QALY, quality-adjusted life year.

from different patient groups were sometimes used to populate the model. For example, in model 1, the diagnostic accuracy of stress ECHO was derived from a study diagnosing significant coronary artery disease, rather than PPCI patients.[31] Despite these potential limitations, our model represents the best available evidence.

Second, for the base-case analysis of model 1, CMR and FFR were treated as reference standards, assuming 100% sensitivity and specificity. We considered identical values for CMR and FFR as a reasonable starting point, while FFR is the reference standard for detecting ischaemia in patients with stable coronary artery disease, there is uncertainty in whether this also applies to a STEMI population with multivessel disease. There is no gold standard test for ischaemia in this population with which to compare. However, it is important to note that varying the sensitivity and specificity in DSAs for model 1 had large impacts on cost-effectiveness results. Similarly, for model 2, reducing the diagnostic accuracy of CMR with standard ECHO compared with standard ECHO doubled the ICER.

Third, resource use was estimated for patients in both models based on standard patient pathways described by clinical experts, as individual patient data from routine sources (eg, Hospital Episode Statistics) were not available at the time of analysis. This may underestimate the variability between individual patients.

Finally, there was uncertainty around the utility decrement of 0.05 for patients with MACE compared with those without MACE.[41] Although this decrement has been used by others, it is not based on primary data.[42] In addition, utility estimates for patients with unobstructed coronary arteries were not available requiring the use of utility estimates from a population with STEMI having PPCI in model 2. We considered these assumptions to be reasonable given (1) quality-of-life using the Short Form-36 was similar for patients with unobstructed coronary arteries and a control group of patients with MI with coronary heart disease[43] and (2) we varied QALYs in the DSAs.

Our cost-effectiveness models have highlighted that the diagnostic accuracies of the tests (CMR and FFR, and CMR) are key drivers of the relative cost-effectiveness of management strategies based on these tests when compared with 'current practice' for patients with multivessel disease and unobstructed coronaries, respectively. In the base-case analyses for patients with multivessel disease, FFR was not cost-effective relative to stress ECHO and was more costly than a CMR strategy. For patients with stable angina, the MR-INFORM trial showed that CMR-guided management is non-inferior for MACE at 1 year compared with invasive angiography and FFR.[44] It is, therefore, concerning that there has been rapid adoption of FFR for detecting ischaemia during our study despite a lack of evidence of benefit over CMR; this may reflect the lack of capacity for CMR nationally. Future research should seek to quantify the relative diagnostic accuracy of CMR and FFR compared with stress ECHO and versus each other. The fact that both CMR and FFR testing are regarded as reference standards in clinical practice and no superior standard is recognised may mean that the cost-effectiveness of CMR compared with FFR can only be tested in a randomised controlled trial, in which the impact of the ischaemia tests on patients' care pathways could be directly observed.

**Author affiliations**
[1]Health Economics Research Centre, Nuffield Department of Population Health, University of Oxford, Oxford, UK
[2]Clinical Trials and Evaluation Unit, Bristol Trials Centre, Bristol Medical School, University of Bristol, Bristol, UK
[3]NIHR Bristol Biomedical Research Centre, University Hospitals Bristol NHS Foundation Trust and University of Bristol, Bristol, UK
[4]Bristol Heart Institute, University Hospitals Bristol NHS Foundation Trust, Bristol, UK
[5]Leeds Institute of Cardiovascular and Metabolic Medicine, University of Leeds, Leeds, UK
[6]University Hospital of Wales, Cardiff, UK

**Acknowledgements** We would like to thank Alice Redfern, who assisted with the literature searches for parameter estimates for the models, and Danielle Bargo, who assisted with the design of the health economic model structures.

**Contributors** EAS designed the study, extracted, analysed and interpreted all the data, critically reviewed and edited the draft article and gave final approval of this version to be published. BD assisted with the interpretation of the data, drafted and revised this article and gave final approval of this version to be published. MP, BCR, CBD and SW conceptualised the study, assisted with its design and the interpretation of data, critically reviewed and edited the draft article and gave final approval of this version to be published. SD assisted with the study design and the interpretation of data, critically reviewed and edited the draft article and gave final approval of this version to be published. JPG and RAA assisted with the interpretation of data, critically reviewed and edited the draft article and gave final approval of this version to be published.

**Funding** This study was funded by the National Institute for Health Research (NIHR) Health Services and Delivery Research Programme (ref 11/2003/58). Maria Pufulete, Barnaby C Reeves, Chiara Bucciarelli-Ducci and the Cardiac MRI Unit are also partly funded through the NIHR Biomedical Research Centre at University Hospitals Bristol NHS Foundation Trust and the University of Bristol.

**Disclaimer** The views expressed are those of the authors and not necessarily those of the NIHR or the Department of Health and Social Care.

**Competing interests** Chiara Bucciarelli-Ducci has received personal fees from Circle Cardiovascular Imaging. Barnaby C Reeves reports former membership of the Health Technology Assessment Commissioning Board (up to 31 March 2016) and Health Technology Assessment Efficient Study Designs Board (October 2014 to December 2014). He also reports current membership of the Health Technology Assessment IP Methods Group and Systematic Reviews Programme Advisory Group (Systematic Reviews NIHR Cochrane Incentive Awards and Systematic Review Advisory Group). Beyond this, all authors have no competing interests, except support from the NIHR grant as detailed in the funding statement.

**Patient consent for publication** Not required.

**Ethics approval** Since the health economic models were populated with data from secondary sources, ethical approval was not required for this study.

**Provenance and peer review** Not commissioned; externally peer reviewed.

**Data sharing statement** No additional data available.

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
