## [Reviewer comments · BMJ Open]

ARTICLE DETAILS

TITLE (PROVISIONAL)	Cardiovascular magnetic resonance in emergency patients with multi-vessel disease or unobstructed coronary arteries: a cost-effectiveness analysis in the United Kingdom
AUTHORS	Stokes, Elizabeth; Doble, Brett; Pufulete, Maria; Reeves, Barnaby; Bucciarelli Ducci, Chiara; Dorman, Stephen; Greenwood, John; Anderson, Richard; Wordsworth, Sarah

VERSION 1 – REVIEW

REVIEWER	Xuanqian Xie Health Quality Ontario, Toronto, Ontario, Canada
REVIEW RETURNED	09-Oct-2018

GENERAL COMMENTS	This study was well done. The model and parameters were introduced transparently. I have a few minor comments for authors' considerations. 1. Please connect editors from BMJ for the structure of Abstract.2. The objective of this study was to identify "key drivers of cost-effectiveness", but the Results of Abstract only reported one factor, the accuracy of test as the single key driver. Based on Results of sensitivity of Table 1 and Table 2, especially Table 2, there are a couple of factors impacted the cost-effectiveness.3. Most economic evaluations report the model inputs in the main text.4. Authors may consider including the commonly used terms (sensitivity, specificity and prevalence) in Appendix A and B, together with the current words (e.g., "Probability of ischaemia test positive and revascularization").5. Authors may introduce the main considerations of using the one-year time-horizon for this study.6. The text in the Figure 1 and 2 were very small. The model can be drawn in another way.7. I think the baseline characteristics, such as "Index angiogram leads to revascularisation of second vessel" and "Uncertainty following index angiogram and need for ischaemia testing", may impact the cost-effectiveness. But I have not found them in the Table 1.8. For the accuracy, authors also can explore the sensitivity and specificity separately.9. Some model inputs were based on the experts' opinion. I would like to understand whether these inputs can substantially impact the cost-effectiveness results, as obviously these inputs were not accurate.10. Authors stated that "While in practice their sensitivity and specificity are likely to be a little less than 100%, there is no gold standard test for ischaemia in STEMI patients with multi-vessel disease with which to compare". In this situation, it is unclear
---

	whether it is best approach to assume 100% sensitivity and 100% specificity. 11. The accuracy of control group (e.g., Stress ECHO and Standard ECHO), or the difference in accuracy between control and intervention, are the key issues. 12. Please connect the clinical experts to confirm the main clinical evidence and key assumptions.
--	---

REVIEWER	Patrick Moore Health Economics Unit, Institute of Applied Health Research, University of Birmingham, UK
REVIEW RETURNED	06-Feb-2019

GENERAL COMMENTS	This study set out to evaluate the drivers of cost effectiveness for Cardiovascular Magnetic Resonance (CMR) when patients activate the Primary Percutaneous Coronary Intervention (PPCI) pathway. The relevant options for evaluation were clearly set out along with the population, setting of participants, timeframe and perspective. The study is generally well executed and explained. I noted the following issues:  1. There appears to be a paucity of evidence on the clinical effectiveness of the three methods: ECHO; CMR and FFR. Surely assessing this should be completed before considering an economic evaluation. 2. The study attempts to identify the key drivers of cost effectiveness (CE), considering a one year period. I appreciate that this is a feasibility study but in the case of cardiovascular disease are there not benefits accrued over a longer time period which would have a potentially large effect on any CE analysis?
--

REVIEWER	Dr Ali Davod Parsa School of Allied Health, Faculty of Health, Education, Medicine and Social Care, Anglia Ruskin University, East Road, Cambridge- CB1 1PT
REVIEW RETURNED	18-Feb-2019

GENERAL COMMENTS	I would not provide the second brief(ARTICLE SUMMARY) than abstract. Reference to the statements "Many patients presenting to hospital with STEMI have multi-vessel disease (40-65%),[2-7] which may require secondary revascularisation of the patients' non-infarcted territories.", this may contradict to the benefits claimed for CMR where said; " The benefits of CMR in other subgroups of PPCI patients, such as those found to have multi-vessel disease, are less clear." the two above statements seem a bit confusing. In the methodology; first, it was said the direct cost and QALYs considered but later is said: "The analysis was conducted from an NHS and personal social services perspective in the UK." If the authors meant societal perspectives then that normally goes beyond the direct cost only. Overall very interesting health economic study. Outcomes benefit healthcare providers such as NHS in more efficient commissioning. Economic models were appropriate and analysis conducted thoroughly. appendices are more informative and some can be included in the assumptions of the methodology section.
--

VERSION 1 – AUTHOR RESPONSE

Reviewers' Comments to Author:

Reviewer: 1

This study was well done. The model and parameters were introduced transparently. I have a few minor comments for authors' considerations.

Response: We thank the reviewer for their positive comments.

1. Please connect editors from BMJ for the structure of Abstract.

Response: In line with the Editorial feedback, we have left the structure of the abstract unchanged.

2. The objective of this study was to identify “key drivers of cost-effectiveness”, but the Results of Abstract only reported one factor, the accuracy of test as the single key driver. Based on Results of sensitivity of Table 1 and Table 2, especially Table 2, there are a couple of factors impacted the cost-effectiveness.

Response: We have limited space in the Abstract to present the factors that influenced cost-effectiveness results. We single out the accuracy of the tests as this was a significant factor in both models, and for model 1, varying the CMR (or stress ECHO) test accuracy were the only sensitivity analyses which altered cost-effectiveness conclusions. In the main text of the manuscript (Sensitivity analysis section of the Results, pages 18-19, and in the second paragraph of the Discussion, pages 19-20), we clearly describe the other factors that influenced cost-effectiveness results.

3. Most economic evaluations report the model inputs in the main text.

Response: We would have liked to include the model inputs in the paper to increase the transparency of our work, however, we felt that this was just too much material to include in the main text. This material amounts to seven pages of the supplementary material – but it is all there and available to readers.

4. Authors may consider including the commonly used terms (sensitivity, specificity and prevalence) in Appendix A and B, together with the current words (e.g., “Probability of ischaemia test positive and revascularization”).

Response: Thank you for this suggestion. We have added the terms sensitivity and specificity to Appendices A and B. We have added prevalence of myocardial infarction to Appendix B for the unobstructed coronary arteries model, but have not added prevalence of ischaemia to Appendix A for multi-vessel disease, since this is not the first branch in the model, and therefore can only be calculated by combining information from several rows of the table. These changes can be found on supplementary material pages 2, 3, and 6.

5. Authors may introduce the main considerations of using the one-year time-horizon for this study.

Response: We have added the following text to the Methods on page 5 to explain our rationale for a one-year time horizon:

“Given the focus on the immediate consequences on clinical management of introducing additional diagnostic testing, and because this is a feasibility study assessing key drivers of cost-effectiveness rather than cost-effectiveness per se, we chose a one-year time horizon. Survival to 12 months under each testing strategy was assessed to determine whether there were differences, which would identify a need to consider a longer time horizon.”

We have added the following text on survival at 12 months to the sensitivity analysis results for Model 1 on page 18:

“Under all testing strategies, the probability of survival to 12 months was 0.99. If a cohort of 1000 patients entered each strategy, on average there would be one more death by 12 months under a stress ECHO strategy compared to CMR/FFR (11 versus 10 deaths). Given that there is no difference in survival, we did not extrapolate findings to a longer time horizon.”

We have added the following text to the sensitivity analysis results for Model 2 on page 19:

“Under both strategies, the probability of survival to 12 months was 0.99. If a cohort of 1000 patients entered each strategy, on average there would be one more death by 12 months under a standard ECHO strategy compared to standard ECHO and CMR (seven versus six deaths). As for Model 1, given that there is no difference in survival, we did not extrapolate findings to a longer time horizon.”

6. The text in the Figure 1 and 2 were very small. The model can be drawn in another way.

Response: We have revised Figures 1 and 2 to make the text larger.

7. I think the baseline characteristics, such as “Index angiogram leads to revascularisation of second vessel” and “Uncertainty following index angiogram and need for ischaemia testing”, may impact the cost-effectiveness. But I have not found them in the Table 1.

Response: The probabilities that:

- the index angiogram leads to revascularisation of a second vessel,
 - there is uncertainty following the index angiogram and a need for ischaemia testing,
 - the index angiogram leads to no revascularisation of a second vessel,
- can be found in the first section of Appendix A (page 2 of the supplementary material). These probabilities are common to each ischaemia testing strategy, so although varying these probabilities affects the mean costs and QALYs under each strategy, it does not alter cost-effectiveness results. This is illustrated in the first sensitivity analysis in Table 1 (page 12), which considered alternative values for these three probabilities. While mean costs and QALYs under each strategy in the sensitivity analysis are different to the base case, the ICER estimates are identical.

8. For the accuracy, authors also can explore the sensitivity and specificity separately.

Response: For each test and each model, we have added sensitivity analyses to explore sensitivity and specificity separately. These additional sensitivity analyses have been added to Appendix C (2a, 2b, 3a, 3b, 4a, 4b) and Appendix D (1a, 1b, 2a, 2b) in the supplementary material (pages 9-11), with results in Tables 1 and 2 of the manuscript, pages 12-13 and 16. Minor changes have been made to

the wording of the results (first paragraph, page 18; and first paragraph, page 19) to reflect these additional sensitivity analyses.

9. Some model inputs were based on the experts' opinion. I would like to understand whether these inputs can substantially impact the cost-effectiveness results, as obviously these inputs were not accurate.

Response: The only probabilities based on expert opinion were the probabilities that

- the index angiogram leads to revascularisation of a second vessel,
- there is uncertainty following the index angiogram and a need for ischaemia testing,
- the index angiogram leads to no revascularisation of a second vessel,

for Model 1 (first section of Appendix A, page 2 of the supplementary material). As described under point 7 above, these probabilities are common to each ischaemia testing strategy, so although varying these probabilities affects the mean costs and QALYs under each strategy, it does not alter cost-effectiveness results. This is illustrated in the first sensitivity analysis in Table 1 (page 12), which considered alternative values for these three probabilities. While mean costs and QALYs under each strategy in the sensitivity analysis are different to the base case, the ICER estimates are identical.

10. Authors stated that "While in practice their sensitivity and specificity are likely to be a little less than 100%, there is no gold standard test for ischaemia in STEMI patients with multi-vessel disease with which to compare". In this situation, it is unclear whether it is best approach to assume 100% sensitivity and 100% specificity.

Response: We discussed this assumption with our clinical collaborators, and in view of the limited options available, the team felt it was reasonable to treat CMR and FFR as reference standards, with assumed 100% sensitivity and specificity. We explored this assumption in sensitivity analyses 2 and 3, which considered the effect on cost-effectiveness results of lower sensitivities and specificities (80%) for CMR, and FFR respectively.

11. The accuracy of control group (e.g., Stress ECHO and Standard ECHO), or the difference in accuracy between control and intervention, are the key issues.

Response: We agree that these are important parameters in the models, which is why a number of the sensitivity analyses considered alternative values for parameters to assess their influence on cost-effectiveness. Sensitivity analyses 2,3 and 4 for Model 1, and 1 and 2 for Model 2, explore test accuracy. (Sensitivity analyses listed in Appendices C and D, pages 9-11 of the supplementary material; and results of the sensitivity analyses reported in Tables 1 and 2, pages 12-14 and 16-17 of the manuscript).

12. Please connect the clinical experts to confirm the main clinical evidence and key assumptions.

Response: Lengthy discussions took place among the team to design the economic models, and then to consider the inputs and assumptions around them. All the material presented has been reviewed by our team, and that includes four clinicians, all of whom are authors on this manuscript. There was limited evidence available on which to base some of the parameter estimates, and therefore we have had to make assumptions; these assumptions are described transparently and we believe our model represents the best available evidence.

Reviewer: 2

This study set out to evaluate the drivers of cost effectiveness for Cardiovascular Magnetic Resonance (CMR) when patients activate the Primary Percutaneous Coronary Intervention (PPCI) pathway. The relevant options for evaluation were clearly set out along with the population, setting of participants, timeframe and perspective. The study is generally well executed and explained.

Response: We thank the reviewer for their positive comments.

I noted the following issues:

1. There appears to be a paucity of evidence on the clinical effectiveness of the three methods: ECHO; CMR and FFR. Surely assessing this should be completed before considering an economic evaluation.

Response: There was limited evidence available from which to draw some of the parameter estimates, however, CMR and FFR are increasingly considered to be standards of care in clinical practice. There is already a risk that, if we delay considering their cost-effectiveness, it is possible that they become part of standard care without ever being evaluated in this way. If evidence around cost-effectiveness is generated, this can influence decisions by commissioners of care and policy makers about whether to specify these tests as standard of care.

2. The study attempts to identify the key drivers of cost effectiveness (CE), considering a one year period. I appreciate that this is a feasibility study but in the case of cardiovascular disease are there not benefits accrued over a longer time period which would have a potentially large effect on any CE analysis?

Response: If there is a difference in mortality between strategies, this would indicate the need to consider a longer time horizon. We have added the following text to the Methods on page 5 to explain our rationale for a one-year time horizon, and to specify when we would consider a longer time horizon:

“Given the focus on the immediate consequences on clinical management of introducing additional diagnostic testing, and because this is a feasibility study assessing key drivers of cost-effectiveness rather than cost-effectiveness per se, we chose a one-year time horizon. Survival to 12 months under each testing strategy was assessed to determine whether there were differences, which would identify a need to consider a longer time horizon.”

We have added the following text on survival at 12 months to the sensitivity analysis results for Model 1 on page 18:

“Under all testing strategies, the probability of survival to 12 months was 0.99. If a cohort of 1000 patients entered each strategy, on average there would be one more death by 12 months under a stress ECHO strategy compared to CMR/FFR (11 versus 10 deaths). Given that there is no difference in survival, we did not extrapolate findings to a longer time horizon.”

We have added the following text to the sensitivity analysis results for Model 2 on page 19:

“Under both strategies, the probability of survival to 12 months was 0.99. If a cohort of 1000 patients entered each strategy, on average there would be one more death by 12 months under a standard ECHO strategy compared to standard ECHO and CMR (seven versus six deaths). As for Model 1, given that there is no difference in survival, we did not extrapolate findings to a longer time horizon.”

Reviewer: 3

I would not provide the second brief(ARTICLE SUMMARY) than abstract.

Response: The Article Summary of strengths and limitations of the study on page 3 is a journal requirement, we have therefore left this in place (with one amendment as requested above by the Editor).

Reference to the statements "Many patients presenting to hospital with STEMI have multi-vessel disease (40-65%),[2-7] which may require secondary revascularisation of the patients' non-infarcted territories.", this may contradict to the benefits claimed for CMR where said; " The benefits of CMR in other subgroups of PPCI patients, such as those found to have multi-vessel disease, are less clear."

the two above statements seem a bit confusing.

Response: The first paragraph of the Introduction on page 4 (which contains the first statement) describes the uncertainty around how to manage patients with STEMI who are found to have multi-vessel disease. For these patients, there is uncertainty about whether or not to revascularise additional vessel(s) in non-infarcted territories in the absence of ischaemia testing. The second paragraph, which contains the second statement, describes CMR ischaemia testing. The benefits of using CMR ischaemia testing to guide patient management are widely accepted for patients with a normal coronary angiogram, but not for other patient subgroups, such as those with multi-vessel disease. We are not saying that CMR testing does not benefit patients with multi-vessel disease, rather that there is a lack of evidence on this question.

In the methodology; first, it was said the direct cost and QALYs considered but later is said: "The analysis was conducted from an NHS and personal social services perspective in the UK." If the authors meant societal perspectives then that normally goes beyond the direct cost only.

Response: The analysis was conducted from an NHS and personal social services perspective, not a societal perspective. We agree that the term 'direct' costs can be ambiguous, we have therefore deleted this word from page 5, and changed:

“... to estimate the direct medical costs and quality-adjusted life years (QALYs)...”
to
“... to estimate the healthcare costs and quality-adjusted life years (QALYs)...”.

Overall very interesting health economic study. Outcomes benefit healthcare providers such as NHS in more efficient commissioning. Economic models were appropriate and analysis conducted thoroughly. appendices are more informative and some can be included in the assumptions of the methodology section.

Response: We thank the reviewer for their positive comments.

VERSION 2 – REVIEW

REVIEWER	Xuanqian Xie Health Quality Ontario, Toronto, Canada
REVIEW RETURNED	06-Apr-2019

GENERAL COMMENTS	It is my pleasure to review the revised manuscript. I do not have further comments.
---

REVIEWER	Patrick Moore University of Birmingham, UK
REVIEW RETURNED	29-Mar-2019

GENERAL COMMENTS	All my previous comments have been appropriately answered.
--